# Feeding Neonates and Infants Prior to Surgery for Congenital Heart Defects: Systematic Review and Meta-Analysis

**DOI:** 10.3390/children9121856

**Published:** 2022-11-29

**Authors:** Douglas Bell, Jessica Suna, Supreet P. Marathe, Gopinath Perumal, Kim S. Betts, Prem Venugopal, Nelson Alphonso

**Affiliations:** 1The Prince Charles Hospital, Brisbane, QLD 4032, Australia; 2Queensland Paediatric Cardiac Service (QPCS), Queensland Children’s Hospital, Brisbane, QLD 4101, Australia; 3School of Clinical Medicine, Children’s Health Queensland Clinical Unit, University of Queensland, Brisbane, QLD 4072, Australia; 4Children’s Health Research Centre, University of Queensland, Brisbane, QLD 4101, Australia; 5UCSF Benioff Children’s Hospital, San Francisco, CA 94158, USA; 6QPCR Collaborators: Janelle Johnson, Tom R Karl, Children’s Health Research Centre, University of Queensland, Brisbane, QLD 4101, Australia

**Keywords:** necrotizing enterocolitis, feeding, neonates, congenital heart disease, cardiac surgery

## Abstract

Background: Necrotising enterocolitis (NEC) is a significant cause of mortality and morbidity in neonates requiring cardiac surgery. Feeding practices vary significantly across institutions and remain controversial. We conducted a systematic review of the literature and a meta-analysis to identify associations between feeding practices and necrotising enterocolitis. Methods: This study was carried out in accordance with the PRISMA guidelines. A literature search was performed in November 2022 using the Cochrane Central Register, Embase, and Pubmed. Two investigators then independently retrieved eligible manuscripts considered suitable for inclusion. Data extracted included gestational age, birth weight, sex, nature of congenital heart lesion, type of operation performed, time on ventilator, ICU stay, hospital stay, post-operative feeding strategy, and complications. The methodological quality was assessed using the Downs and Black score for all randomised control trials and observational studies. Results: The initial search yielded 92 studies. After removing duplicates, there were 85 abstracts remaining. After excluding ineligible studies, 8 studies were included for the meta-analysis. There was no significant risk of NEC associated with pre-operative feeding [OR = 1.22 (95% CI 0.77,1.92)] or umbilical artery catheter placement [OR = 0.91 (95% CI 0.44, 1.89)] and neither outcome exhibited heterogeneity [I^2^ = 8% and 0%, respectively]. There was a significant association between HLHS and NEC [OR = 2.56 (95% CI 1.56, 4.19)] as well as prematurity and NEC [OR 3.34 (95% CI 1.94, 5.75)] and neither outcome exhibited heterogeneity [I^2^ = 0% and 0%, respectively]. Conclusions: There was no association between NEC and pre-operative feeding status in neonates awaiting cardiac surgery. Pre-operative feeding status was not associated with prolonged hospital stay or need for tube assisted feeding at discharge. HLHS and prematurity were associated with increased incidence of NEC.

## 1. Introduction

Necrotizing enterocolitis (NEC) has a prevalence of approximately 5% in neonates born with congenital heart disease (CHD), compared with a prevalence of 0.2% in the general newborn population [1]. The incidence of NEC is even higher in neonates born with left heart obstructive lesions, in whom the incidence may be as high as 13% [2]. The risk of mortality following NEC in neonates with CHD ranges between 25 and 50% [3]. In neonates with CHD, blood flow to the gastrointestinal (GI) tract may be suboptimal. Many such babies have a lower haemoglobin oxygen saturation as compared to the general newborn population. Furthermore, diastolic steal and flow reversal in the abdominal aorta may be problematic in those with a large ductus arteriosus [4]. Prostaglandins used to maintain ductal patency may reduce gastric acid secretion and prolong transit times of enteral feeds, which can facilitate bacterial translocation and promote the development of NEC. The need for a prostaglandin infusion in those with duct-dependent circulations has been associated with a higher incidence of NEC [5]. These concerns have resulted in a reluctance to feed neonates awaiting cardiac surgery at many institutions [6]. In addition, use of inotropes, prostaglandins for ductal patency, and umbilical artery catheters are relative contraindications to pre-operative feeding at some institutions [7].

On the other hand, some reports suggest that there are deleterious effects from withholding feeding prior to cardiac surgery. Feeding restrictions may compromise metabolic reserves which are especially important for neonates who develop a profound systemic inflammatory response to cardiopulmonary bypass, as compared to older children and adults [8]. Restricted feeding has been associated with longer stay following cardiac surgery in neonates [9]. Delayed feeding may also contribute to cellular atrophy and increased permeability of the gastrointestinal tract [10]. Worse neurodevelopmental outcomes have been associated with restricted feeding practices in neonates [11]. It has been suggested that pre-operative feeding may improve post-operative feeding tolerance [12]. Due to these competing risks and conflicting data, feeding practices vary among surgeons, cardiologists, intensivists, and institutions. The use of pre-operative feeding ranges from 29% to 79% across institutions [6]. In addition, there are no published guidelines for feeding practices in neonates awaiting cardiac surgery.

The primary objective of this systematic review was to determine if pre-operative enteral feeding is associated with NEC in neonates and infants undergoing cardiac surgery. The secondary objectives of this study were:

(A) To identify additional risk factors for NEC; specifically, hypoplastic left heart syndrome (HLHS), use of umbilical artery catheters and prematurity

(B) To identify adverse events associated with pre-operative feeding practices including length of hospital stay and need for tube assisted feeding at the time of discharge.

## 2. Materials and Methods

### 2.1. Search Strategy

This systematic review and meta-analysis was carried out in accordance with the PRISMA guidelines [13]. Using the Cochrane Central Register, Embase, and Pubmed, a literature search was performed in November 2022 using the following terms: ‘enteral feeding’ [MeSH Terms] AND ‘cardiac surgery’ OR ‘congenital heart disease’. [MeSH Terms] AND ‘necrotising enterocolitis’ [MeSH Terms] AND ‘neonates’ [MeSH Terms] OR ‘infants’ [MeSH Terms]. Two investigators (DB and GP) then independently retrieved eligible manuscripts considered suitable for inclusion. Both investigators independently examined the design, patient population and interventions in each manuscript. The reference lists of included papers and relevant systematic reviews were screened, and electronic author and citation tracking was performed to identify relevant publications not identified at the initial search strategy. Where crucial data were missing from available manuscripts, authors were contacted to provide the relevant data. Studies were limited to English language and human subjects only. There was no limit on study eligibility by study design.

### 2.2. Selection Criteria

The population included children less than one year of age undergoing cardiac surgery. The intervention studied was pre-operative enteral feeding. Various enteral feeding regimens were compared including no feeding, ‘trophic’ feeding (<20 mL/kg/day) and full feeds. The primary outcome of interest was NEC as defined by the Bell criteria stage 2 or greater [14]. Secondary outcomes included mortality, length of ventilator time, duration of hospital stay and need for tube assisted feeding post-operatively. Studies which did not focus on the human CHD population in the first year of life were excluded. Lastly, where a study presented more than one set of effect estimates, we used the estimates from the final (fully adjusted) model. When no model was used, we derived the effect estimate from the raw numbers.

### 2.3. Data Extraction

Authorship, year of publication, type of publication, study design, length of follow-up, patient population and sample size were examined. Data extracted included gestational age, birth weight, sex, nature of congenital heart lesion, type of operation performed, ventilation time, ICU and hospital length of stay, post-operative feeding strategy and complications. The methodological quality was assessed by two independent investigators (DB and GP) using the Downs and Black score [15] [total score from 0 (poor) to 29 (excellent)] for all randomised control trials and observational studies. Disagreements regarding scores were resolved by consensus and by consulting senior authors if required.

### 2.4. Statistical Analysis

Our meta-analyses included four binary outcome variables which were assessed by at least two studies. Effect estimates were entered into the meta-analysis as the (log) odds ratio (OR) and standard error. For studies without reported ORs, we used the Practical Meta-analysis Effect Size Calculator, to convert given effect estimates to ORs, or calculated the OR and 95% CI by hand if raw numbers were given and no multivariable analysis was performed [16,17]. When calculating by hand, the Haldane-Anscombe correction was applied, in which 0.5 is added to each cell if one of the cells has a 0. Next, separate random-effects meta-analyses were performed for each of the four outcomes using restricted maximum likelihood estimation (REML), with the extent of heterogeneity calculated using the I^2^ statistic and Cochran’s Q at a 95% level of error [18,19]. Visual assessments of publication bias were made by inspecting funnel plots of study log OR versus study standard error (S.E.) of the log OR for asymmetry and statistically by using the Egger’s test [20].

## 3. Results

### 3.1. Selection and Characteristics of Included Studies

The initial search using the terms specified above, yielded 92 studies. After removing duplicates, there were 85 abstracts remaining, which were subsequently reviewed. Of these, 6 articles were on a non-related topic, 2 used non-human subjects, 9 were review articles, 5 were conference abstracts and 1 article was unable to be accessed in English. After removing these studies, the remaining 62 articles were included for the full manuscript review. A total of 54 studies were removed during this process. Two studies did not report the incidence of NEC. Nine studies did not specify feeding regimens, 21 studies included groups other than those with congenital heart disease and 21 studies did not specifically identify pre-operative feeding practices. One further study had zero incidence of NEC in both groups and so could not be included for meta-analysis, leaving eight studies (Figure 1, Table 1) eligible for inclusion in the meta-analysis [3 case–control studies [5,21,22], 4 cohort studies [23,24,25,26] and one randomised control trial [27]]. The randomized controlled trial compared those who had not been fed pre-operatively with those who had been given oral feeds and found no difference in the incidence of NEC in the *nil per os* (NPO) group (8%) vs. the trophic feeding group (14%) (*p* = 0.9). Four of the studies were prospective and listed NEC as the outcome and identified associated risk factors. The remaining three were retrospective studies. There was no association between pre-operative feeding and the development of NEC.

### 3.2. Pre-Operative Feeding and NEC

Overall, there was no significant risk of NEC associated with pre-operative feeding, and no indication of study heterogeneity [Q = 8.14, df = 7, *p* = 0.32, I^2^ = 8%]. The pooled odds ratio for pre-operative feeding associated with NEC was 1.22 (95% CI, 0.77 1.92; *p* = 0.39) [Figure 2a]. Further, neither the funnel plot [Figure 2b] nor Egger’s test [*p* = 0.769] indicated significant publication bias. Because one of the included studies was an RCT, we re-ran the meta-analyses excluding this study to see if the results changed. We found that the results did not change substantively [1.19 (95% CI, 0.74 1.93; *p* = 0.473); Q = 8.01, df = 6, *p* = 0.238, I^2^ = 8%].

### 3.3. Association of Additional Risk Factors and NEC

Five studies [5,23,24,25,26] examined the impact of HLHS on NEC [Figure 3a]. There was a significant association between HLHS and NEC OR = 2.56 (95% CI, 1.56, 4.19) and no indication of study heterogeneity [Q = 2.46, df = 4, *p* = 0.65, I^2^ = 0%]. The funnel plot [Figure 3b] and Egger’s test [*p* = 0.637] did not indicate significant publication bias.

Three studies examined an association between umbilical artery catheter placement and NEC [5,23,24]. There was no relationship between placement of an umbilical artery catheter and NEC, OR = 0.91 (95% CI, 0.44, 1.89) [Figure 4a], and no indication of study heterogeneity [Q = 0.42, df = 2, *p* = 0.81, I^2^ = 0%] or publication bias [Figure 4b; Egger’s test *p* = 0.637].

Four studies examined the association between prematurity and NEC [5,24,25,26]. There was a significant association between prematurity and NEC OR = 3.34 (95% CI, 1.94, 5.75) [Figure 5a]. For this analysis there was no indication of study heterogeneity [Q = 1.41, df = 3, *p* = 0.70, I^2^ = 0%] or publication bias [Figure 5b; Egger’s test *p* = 0.581].

### 3.4. Additional Outcomes Following Pre-Operative Feeding

Two studies [24,27] assessed the relationship between preoperative feeding and need for tube assisted feeding post-operatively, with each study finding no association individually [Scahill: OR = 0.90 (0.43, 1.83); Zyblewski: OR = 1.10 (0.37, 3.25)], and with no association overall [OR = 0.95 (0.52, 1.74), *I^2^ = 0.0*].

Two studies [24,27] compared the difference in hospital length of stay by feeding versus NPO, with Scahill et al. finding a mean increase of six days among those fed (95% CI = −8.56, 20.56) while Zyblewski et al. found that pre-operative feeding was associated with a ten-day decrease in hospital length of stay (95% CI= −27.27, 7.27). The meta-analysis result was non-significant (mean diff = −1.30 (−16.92, 14.32; *I^2^ = 48%-estimated as a random effect)*.

## 4. Discussion

Our analysis found no evidence of an association between pre-operative feeding and NEC in patients with CHD. We also found no association between with-holding feeds pre-operatively and need for post-operative tube assisted feeding or prolonged hospital stay and no relationship between placement of an umbilical artery catheter and NEC.

However, we identified an association between NEC and prematurity, and the diagnosis of HLHS. Impaired mesenteric artery flow in those who undergo the Norwood procedure has been proposed as a possible explanation for this association [28]. Prematurity and a diagnosis of HLHS have been previously identified as risk factors for NEC and findings from our meta-analysis agree with these findings [24,25,27,28].

There was no association found between feeding practices and the need for post-operative feeding tube or duration of hospital stay. We were unable to include prostaglandin dependence as a variable for meta-analysis because of the variation in reporting prostaglandin use across the studies. Factors such as ventilation time and genetic syndromes, which were not included in this meta-analysis, have also been associated with feeding difficulties in neonates requiring cardiac surgery (9,10). McKean and colleagues found that feeding difficulties were associated with the need for assisted feeding pre-operatively (OR = 4.4, *p* = 0.03), genetic syndromes (*p* < 0.0001) and the need for a palliative procedure before a biventricular repair (OR = 5.1, *p* = 0.02) [11].

These findings support the relative safety of oral feeding for neonates awaiting cardiac surgery. An analysis of the paediatric critical care consortium registry revealed significant variation in feeding practices for those born with CHD [4]. There remains very little evidence on which to base feeding decisions, and the optimal feeding strategy for infants with CHD remains unknown.

### Limitations

The number of patients in included studies was small, and the power to detect an association is therefore limited. The current evidence is insufficient to draw substantial conclusions regarding the most appropriate pre-operative feeding practices for neonates requiring cardiac surgery. Although NEC is overall a rare entity, the associated mortality and morbidity are high. The decision to feed or withhold feeding from a neonate is multifactorial. Clinician preference forms a significant part of the decision process, which creates very heterogenous comparison groups when pooling patients from different studies. This limits our ability to draw strong conclusions. Increasingly, feeding protocols are often implemented in neonates undergoing cardiac surgery. Such protocols include a series of decision matrices which dictate feed volume and speed based on the patient’s clinical status. The limited number and retrospective nature of studies limit the assessment of a binary decision process (feeding vs. not feeding) and make it impossible to assess to effectiveness of more complex feeding decision tools including the volume of feed. Causative links between feeding outcomes and oral status are yet to be established and hence randomised controlled trials are required.

The present study justifies future studies including randomised controlled trials on the impact of feeding neonates who require cardiac surgery. Future research should also focus on developing strategies to reduce the risk of NEC in high-risk groups such as premature neonates and those with a diagnosis of HLHS.

## 5. Conclusions

Findings from our systematic review and meta-analysis identified no association between pre-operative feeding status and NEC in neonates awaiting cardiac surgery. Feeding status was not associated with prolonged hospital stay, placement of an umbilical artery catheter or need for tube assisted feeding at discharge. HLHS and prematurity were associated with NEC.

## Figures and Tables

**Figure 1 children-09-01856-f001:**
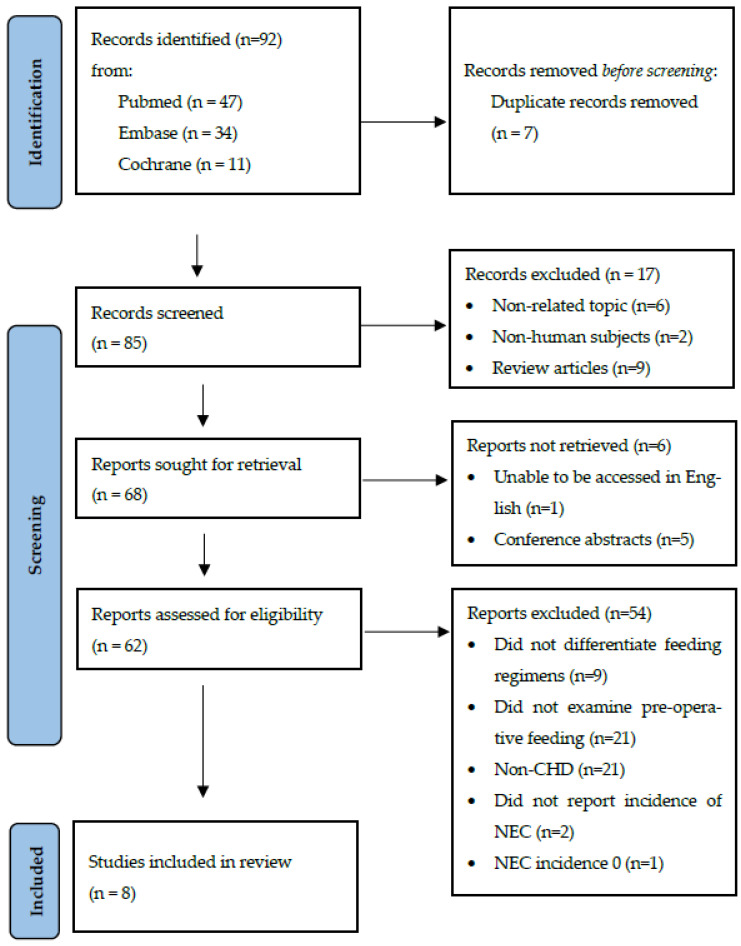
PRISMA flow diagram of search strategy. Terms: [Enteral feeding] AND [Cardiac Surgery] OR [congenital heart disease] AND [Necrotising Enterocolitis] AND [neonates OR infants].

**Figure 2 children-09-01856-f002:**
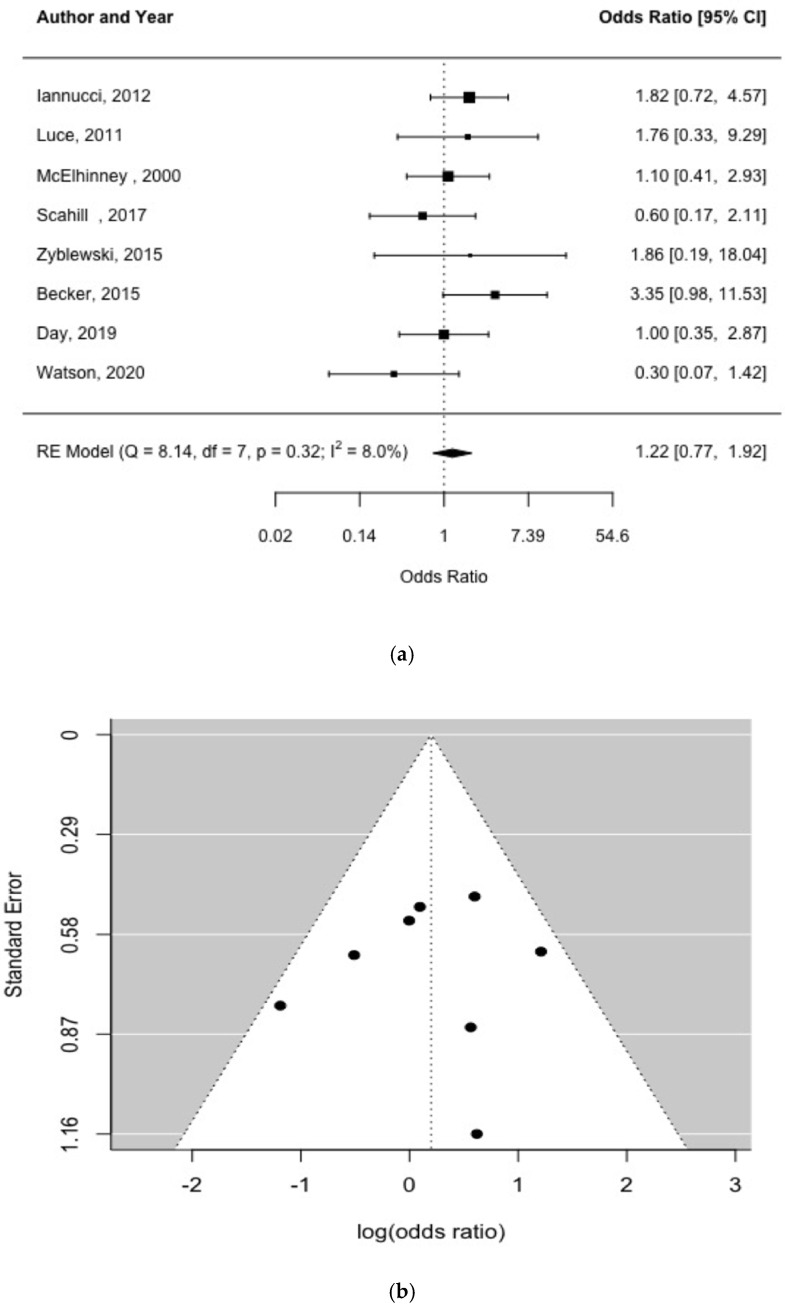
Forest plot (**a**) and funnel plot (**b**) of study effect estimates for preoperative feeding and NEC [shown as odds ratios and log (odds ratios), respectively] [5,21,22,23,24,25,26,27]. Egger’s test did not indicate funnel plot asymmetry [*p* = 0.769].

**Figure 3 children-09-01856-f003:**
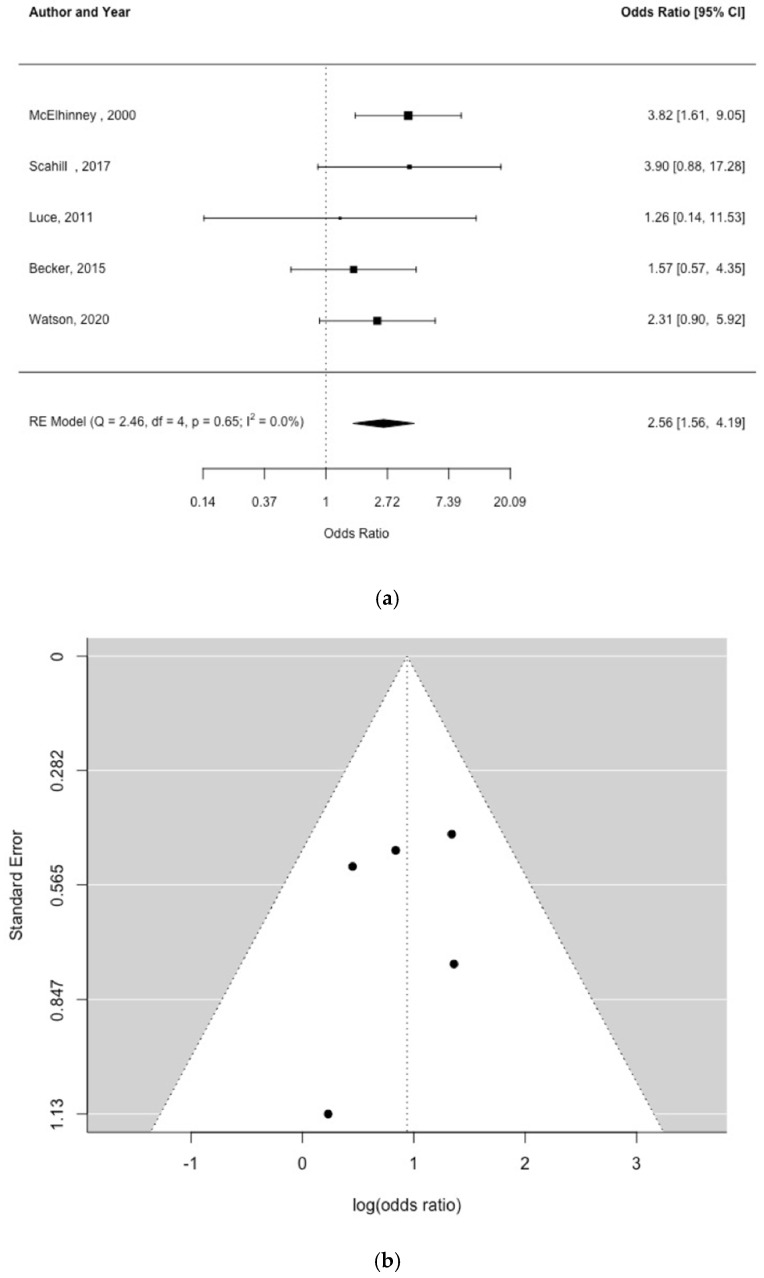
Forest plot (**a**) and funnel plot (**b**) of study effect estimates for hypoplastic left heart syndrome and NEC [shown as odds ratios and log (odds ratios), respectively] [5,23,24,25,26]. Egger’s test did not indicate funnel plot asymmetry [*p* = 0.633].

**Figure 4 children-09-01856-f004:**
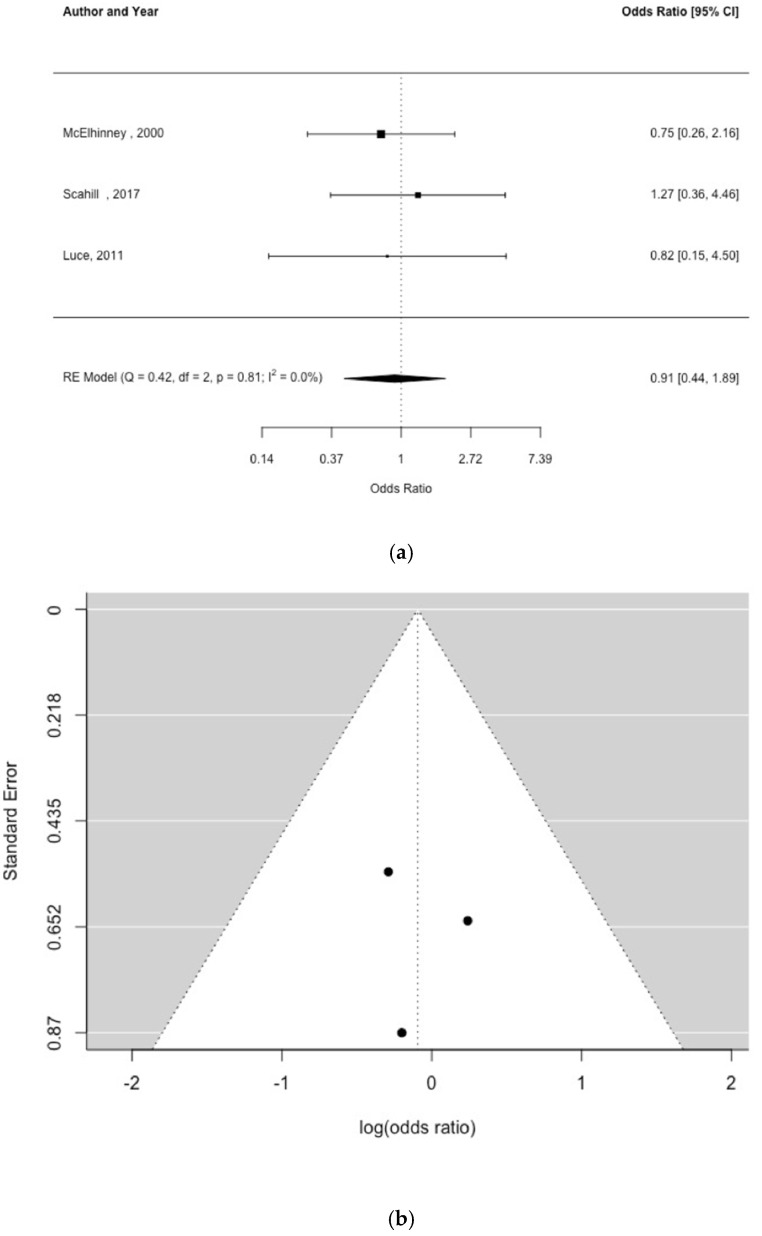
Forest plot (**a**) and funnel plot (**b**) of study effect estimates for umbilical catheter and NEC [shown as odds ratios and log (odds ratios), respectively] [5,23,24]. Egger’s test did not indicate funnel plot asymmetry [*p* = 0.914].

**Figure 5 children-09-01856-f005:**
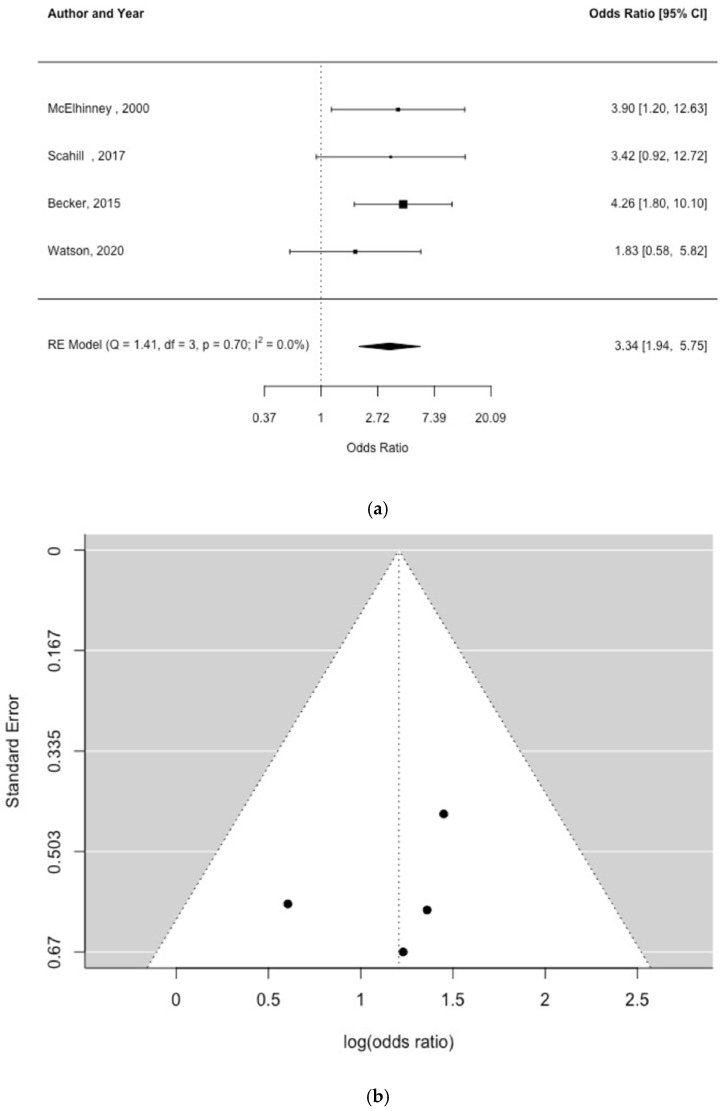
Forest plot (**a**) and funnel plot (**b**) of study effect estimates for premature birth and NEC [shown as odds ratios and log (odds ratios), respectively] [5,24,25,26]. Egger’s test did not indicate funnel plot asymmetry [*p* = 0.581].

**Table 1 children-09-01856-t001:** Eight studies included in the meta-analysis in chronological order.

Author	Year	Study Design	Cohort Size	Comparison	Bell Stage	Odds Ratio for NEC Without Pre-Operative Feeding	Mean Quality Score (Down and Black)
McElhinney et al. [5]	2000	Case control study	643 in inception cohort 21 cases of NEC matched with 70 controls from the inception cohort	NEC vs. non NEC	II or above	1.10 (0.41, 2.92)	20
Luce et al. [23]	2011	Retrospective chart review	73 consecutive neonates	NEC vs. non NEC	II or above	1.76 (0.33, 9.40)	19
Iannucci et al. [22]	2012	Retrospective case–control study.	81 27 NEC matched to 54 controls	NEC to matched case–controls (NEC vs. non NEC)	II or above	1.82 (0.71, 4.62)	18
Becker et al. [25]	2015	Retrospective cohort study	6710 infants	NEC vs. non NEC	II or above	1.08 (0.38, 11.7)	18
Zyblewski et al. [27]	2015	Randomized control trial	27 term born neonates 13 randomized to NBM 14 randomised to oral trophic feeds (10 mL/kg/day)	Feeding vs. not feeding	II or above	NO NEC	22
Scahill et al. [24]	2017	Retrospective cohort study	131 consecutive neonates undergoing surgery for congenital heart disease	NEC vs. non NEC	II or above	0.6 (0.1, 2.1)	17
Day et al. [21]	2019	Case control study	177 consecutive patients diagnosed with hypoplastic left heart syndrome, coarctation of the aorta, pulmonary atresia, or transposition of the great arteries	NEC vs. non NEC	II or III	0.30 (0.02, 5.75)	20
Watson et al. [26]	2020	Prospective cohort study	103 consecutive infants <120 days of age undergoing cardiothoracic surgery with cardiopulmonary bypass	Feeding vs. not feeding	I or II	1.00 (0.35, 2.86)	20

## Data Availability

Not applicable.

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
