# Peer review of "Feeding Neonates and Infants Prior to Surgery for Congenital Heart Defects: Systematic Review and Meta-Analysis"

_children, 2022, doi:10.3390/children9121856_

Round 1

Reviewer 1 Report

Thank you for asking to review Bell et al’s systematic review and meta-analysis. The review aimed to assess a controversial topic of feeding vs no feeding for infants with congenital heart disease prior to their cardiac surgery. It is an important question to be answered.

My comments:

-          Search was done in Dec 2020. It will need to be updated given the potential more publications by now. There seems to be more articles such as PMID: 33009358. If these are excluded the authors need to mention the reason to exclude such studies.

-          Abstract: I suggest to add the effect estimates for NEC and UACs along with the heterogeneity. NEC is the primary outcome for this meta-analysis.

-          Search strategy: I suggest to include “congenital heart disease” OR “cardiac surgery” instead of “cardiac surgery” alone.

-          Figure 1: please use PRISMA structure.

-          Line 194:  “authors” does not fit the sentence. Please review.

-          The meta-analysis combined RCTs and observational studies. Although this can be done, the effect estimates of the analysis should be presented for the RCTs (although only one available) and observational studies separately before combining them. This can be done within the same figure.  

-          The studies characteristics table has feeding vs no feeding but did not specify whether the feeding was trophic or more than trophic. I would suggest adding this important information. Also, a sensitivity analysis based on the feed volume is suggested here.

-          The authors mentioned the need for 3 studies to run the analysis. This role needs to be referenced. Usually 2 studies are enough to run the analysis.  

Author Response

Thank you for asking to review Bell et al’s systematic review and meta-analysis. The review aimed to assess a controversial topic of feeding vs no feeding for infants with congenital heart disease prior to their cardiac surgery. It is an important question to be answered.

My comments:

  1. Search was done in Dec 2020. It will need to be updated given the potential more publications by now. There seems to be more articles such as PMID: 33009358. If these are excluded the authors need to mention the reason to exclude such studies.

Response: We have now conducted a new search to include studies till October 2022. None of the articles in the new search could be included in the meta-analysis. We have updated the PRISMA chart to reflect the new numbers including the reasons for exclusion. Specifically examining the study with PMID:33009358 (Kataria-Hale et al.), the study could not be included as it addresses patients with any kind of intervention (including trans-catheter interventions). The study does not explicitly mention which patients had cardiac surgery, thereby not fulfilling the inclusion criteria of our meta-analysis.

Changes: New PRISMA chart

  1. Abstract: I suggest adding the effect estimates for NEC and UACs along with the heterogeneity. NEC is the primary outcome for this meta-analysis.

Response: We have now included these in our abstract.

Changes: We have now included these in our abstract (abstract, paragraph 3).

  1. Search strategy: I suggest to include “congenital heart disease” OR “cardiac surgery” instead of “cardiac surgery” alone.

Response: We have conducted the new search included in Point 1 including the above-mentioned terms

Changes: Updated list of search terms

  1. Figure 1: please use PRISMA structure

Response: We have reported the results in the PRISMA format (page 4, figure 1)

  1. Line 194: “authors” does not fit the sentence. Please review

Response: We have corrected this error.

  1. The meta-analysis combined RCTs and observational studies. Although this can be done, the effect estimates of the analysis should be presented for the RCTs (although only one available) and observational studies separately before combining them. This can be done within the same figure.

Response: As there was only one RCT, we have re-run the feeding meta-analysis excluding this study and included the results in the paper.

Changes: We have added the results from excluding the RCT in section 3.2 lines 190-193.

  1. The studies characteristics table has feeding vs no feeding but did not specify whether the feeding was trophic or more than trophic. I would suggest adding this important information. Also, a sensitivity analysis based on the feed volume is suggested here.

Response: While stratification of feed volumes will be ideal, unfortunately, all studies do not include this variable. Also, different studies have defined these terms differently. Consequently, we will be unable to do such an analysis. We have elaborated this aspect in the limitations section.

Changes: None

  1. The authors mentioned the need for 3 studies to run the analysis. This role needs to be referenced. Usually, 2 studies are enough to run the analysis.

Response: The reviewer is correct, this was a typo, it was meant to say two studies.

Changes: This has been changed from “three” to “two”.

Reviewer 2 Report

The authors present an interesting systematic review of the association between pre-op feeding and NEC in congenital heart disease surgery patients. This is an area of much interest in the care of this patient group and the authors rightly point out the wide practice variation. The paper is well written and concise.

The methodology of the review appears sound. The limitations related to the analysis are mainly due to the small sample sizes and study heterogeneity of the studies included in the meta-analysis which the authors address in the limitations.

The results are clearly reported with helpful figures. It would be helpful to include a little more information about the patient population for each study to get an idea of the mix of diagnoses if that information is available. You did this in Table 1 for the Day et al study. It does not need to be a comprehensive list, but at least an idea of the range of diagnoses or procedures included in the study. It would be specifically helpful to point out which included HLHS patients, although this can be gleaned from the Figure 3 results.

The discussion appropriately reviews the results and compares them to previous results. Limitations are well described.

Author Response

The authors present an interesting systematic review of the association between pre-op feeding and NEC in congenital heart disease surgery patients. This is an area of much interest in the care of this patient group and the authors rightly point out the wide practice variation. The paper is well written and concise.

The methodology of the review appears sound. The limitations related to the analysis are mainly due to the small sample sizes and study heterogeneity of the studies included in the meta-analysis which the authors address in the limitations.

The results are clearly reported with helpful figures. It would be helpful to include a little more information about the patient population for each study to get an idea of the mix of diagnoses if that information is available. You did this in Table 1 for the Day et al study. It does not need to be a comprehensive list, but at least an idea of the range of diagnoses or procedures included in the study. It would be specifically helpful to point out which included HLHS patients, although this can be gleaned from the Figure 3 results.

The discussion appropriately reviews the results and compares them to previous results. Limitations are well described.

Response: Thank you for your kind comments

Changes: None

Round 2

Reviewer 1 Report

Responses are acceptable